# Effect of Thermal Pretreatment on the Physiochemical Properties and Stability of Pumpkin Seed Milk

**DOI:** 10.3390/foods12051056

**Published:** 2023-03-02

**Authors:** Min Yu, Mengyao Peng, Ronghua Chen, Jingjing Chen

**Affiliations:** 1State Key Laboratory of Food Science and Technology, Jiangnan University, Wuxi 214122, China; 2International Joint Laboratory on Food Safety, Jiangnan University, Wuxi 214122, China

**Keywords:** pumpkin seed milk, viscosity, particle size, zeta potential, structural properties

## Abstract

During the production of plant-based milk, thermal treatment of raw materials is an important processing method to improve the physicochemical and nutritional quality of the final products. The objective of this study was to examine the impact of thermal processing on the physiochemical properties and stability of pumpkin seed (*Cucurbita pepo* L.) milk. Raw pumpkin seeds were roasted at different temperatures (120 °C, 160 °C, and 200 °C), and then processed into milk using a high-pressure homogenizer. The study analyzed the microstructure, viscosity, particle size, physical stability, centrifugal stability, salt concentration, heat treatment, freeze–thaw cycle, and environment stress stability of the resulting pumpkin seed milk (PSM120, PSM160, PSM200). Our results showed that the microstructure of pumpkin seeds was loose and porous, forming a network structure because of roasting. As the roasting temperature increased, the particle size of pumpkin seed milk decreased, with PSM200 showing the smallest at 210.99 nm, while the viscosity and physical stability improved. No stratification was observed for PSM200 within 30 days. The centrifugal precipitation rate decreased, with PSM200 showing the lowest rate at 2.29%. At the same time, roasting enhanced the stability of the pumpkin seed milk in the changes in ion concentration, freeze–thaw, and heating treatment. The results of this study suggested that thermal processing was an important factor in improving the quality of pumpkin seed milk.

## 1. Introduction

In recent years, consumers have shown an increased interest in plant-based milks as a healthier alternative to traditional animal milk. The reason why consumers prefer to consume alternative milks is because they do not contain cholesterol and can reduce cardiovascular disease risk and allergic risk [1,2]. Besides, since plant-based milk does not contain lactose, there are no concerns about lactose intolerance, which is otherwise a serious problem in milk [3]. Lastly, plant-based milk is thought to be more environmentally friendly due to the lower greenhouse gas emissions associated with plant farming compared to animal farming [4].

At present, there are many varieties of plant-based milk on the market, such as almond, soy, oat, and cashew milk. Almond milk is low in calories and relatively high in calcium, but low in protein (1.44%). Soy milk contains almost twice as much folic acid and vitamin B12 as cow’s milk [5]. Rincon had developed a new plant-based milk based on chickpeas and coconut that had a higher protein calcium fat content than oat milk [6]. However, stability remains a focus of plant milk research. There are many factors affecting the stability of plant-based milk, including protein–oil ratio processing methods, pH conditions, ion strength, environmental temperature, etc. [7,8]. Pan et al. [9] hydrolyzed rice protein with different proteases (neutral enzyme trypsin and alkaline protease) and found that the trypsin hydrolysate showed high stability. The stability of plant-based milk mixed with quinoa and lentil protein was also improved by regulating protein–protein interactions and the interaction between protein and quinoa starch [10]. These findings demonstrate that further research into improving the stability of plant-based milk is needed.

Heat treatment is a critical step in the production and processing of milk alternatives [11]. Roasting, in particular, is one of the heat treatment processing methods. It uses the heating principle to make the product evenly cooked and improves the digestibility and sensory quality of food nutrition by studying the transformation of the food matrix with an ideal structure [12,13]. Although heat processing can degrade and aggregate proteins, resulting in a decreased protein content, proper heat processing can improve functional properties by increasing protease contact sites [14]. Moreover, after heat treatment, the polypeptide chain will unfold, and the sulfhydryl group and hydrophobic side chain inside the molecule will be exposed, affecting the functional properties of the protein. High temperatures can induce protein interactions, leading to protein aggregation and precipitation [15]. Dissociation and aggregation caused by heating have been extensively studied in seeds such as soybeans, oats, and kidney beans [16]. The increase in temperature during heat processing will promote the degradation of polysaccharide structure, reduce the particle size of substances, and improve the stability and storage period of food [17]. Jin et al. [18], in their experiments on the effects of roasting on the physical and chemical properties of sesame paste, pointed out that roasting increased the texture and rheology of the paste, reduced the particle size of sesame paste, and improved its storage stability.

Pumpkin seeds are a rich source of high-quality plant protein and oil. In addition to common nutrients such as carbohydrates and proteins, pumpkin seeds also contain vitamins (B1, B2, E, etc.), carotenoids, squalene, phytosterols, cucurbitacin, and phenolic compounds [19]. Pumpkin seed extract has been shown to have several health benefits, including the prevention of prostate cancer and urinary system diseases [20], and the improvement of symptoms associated with hypertension and diabetes [21]. Mixing pumpkin seed milk and camel milk can change the chemical properties, antioxidant, viscosity, and sensory properties of fermented camel milk, while increasing the phenolic components and antioxidant dietary fiber, thus improving the nutritional value and health benefits [22]. Kuru et al. [23] studied the optimization of the plant milk mixing design. After sunflower seeds and pumpkin seeds were combined, both the dry matter and ash contents of samples had increased. The content of total milk phenol and DPPH free radical scavenging activity were dominated by the sunflower seed ratio. The hazelnut ratio had positive effects on the protein content, whiteness index, serum stability, and sensory properties. The purpose of this experiment was to study the effects of roasting materials on the physicochemical properties and stability of pumpkin seed milk. 

## 2. Materials and Methods

### 2.1. Material

Pumpkin seeds (*Cucurbita pepo* L.) were provided by Sanxin Company (Ürümqi, Xinjiang, China). Reagents used in this study were of analytical grade unless otherwise specified. N-hexane was purchased from InnoChem Science and Technology Co., Ltd. (Beijing, China). Disodium hydrogen phosphate dodecahydrate, sodium phosphate dibasic dihydrate, disodium tetraborate decahydrate, O-phthalic aldehyde (OPA), dithiothreitol (DTT), serine, folin-phenol, sodium carbonate, trichloroacetic acid (TCA), absolute ethyl alcohol, sodium chloride, sodium hydroxide, calcium chloride, potassium chloride, sodium bicarbonate, and magnesium chloride, were purchased from Sinopharm Chemical Reagent Co., Ltd. (Shanghai, China). Coomassie Blue Fast Staining Solution G250, Bovine Serum Albumin (BSA), and Phosphate-Buffered Saline (PBS) were purchased from Beyotime Biotechnology Co., Ltd. (Shanghai, China).

### 2.2. Preparation of Pumpkin Seed Milk

Three equal masses of pumpkin seeds were roasted at 120 °C, 160 °C, and 200 °C for 10 min, respectively, in an oven (NB-HM3810, Panasonic Manufacturing Xiamen Co., Ltd., Xiamen, China). Then, the roasted pumpkin seeds were soaked in distilled water with a ratio of 1:3 (*w*/*w*) at 4 °C for 20 h [24]. After soaking, the seed coats were carefully removed. The soaking water was drained and the residual water on the naked seeds was removed with a kitchen paper towel. Then, the seeds were grounded with water at the weight ratio of 1:8 (*w*/*v*) in a blender (HR2101, Philips, Amsterdam, The Netherlands) for 3 min. The slurry was filtered with eight layers of cheesecloth. Then, the filtrate was homogenized twice with a high-pressure homogenizer (JHG-54-P100, GEA Mechanical Equipment Italia S.P.A., Parma, Italy) at 40 MPa. Pumpkin seed milk (PSM) prepared with seeds at different roasting temperatures was obtained with the same method and named: RAW, PSM120, PSM160, and PSM200.

### 2.3. Characterization of Pumpkin Seed Milk

#### 2.3.1. Microstructure of PSMs

##### Confocal Laser Scanning Microscopy (CLSM) of PSMs

The microstructure of roasted pumpkin seed milk droplets was observed using a confocal laser microscope (LSM710, Carl Zeiss AG, Oberkochen, Germany) method reported by Zhong et al. [25]. Briefly, 0.1 mL of premixed Nile red and Nile blue A (0.1% in isopropyl alcohol) was added into 2 mL of pumpkin seed milk and placed in the dark for 30 min. Then, the mixture was dropped on a microscope slide and observed with the confocal laser microscope. The fluorescence of the dyes was excited at the wavelength of 488 nm for Nile red and 633 nm for Nile blue A.

##### Microstructure of PSMs

Pumpkin seed milk was freeze-dried. The freeze-dried solid samples were evenly spread on the sample table with conductive adhesive and treated with gold spraying. Imaging was observed under an accelerated voltage of 20 kV with a scanning electron microscope (TM3030, Koki Holdings Co. Ltd., Tokyo, Japan).

#### 2.3.2. Particle Size Distribution of PSMs

Each PSM sample was diluted 300 times with deionized water. The particle size distribution of the sample was determined using a multi-angle particle size and high sensitivity ζ-potential analyzer (Nano Brook Omni, Malvern Instruments, Malvern, Worcestershire, UK) after 2 h and 7 days of storage (20 °C) [26].

#### 2.3.3. Rheological Properties of PSMs

A rheometer (DHR-3, Kinexus, Malvern, Worcestershire, UK) was used to determine the rheological properties of PSMs [27]. The instrument was equipped with a cone plate. A plate gap of 0.1 mm and plate truncation of 1° was maintained throughout the experiment. Milk samples (2 mL) were loaded on the plate at a temperature of 25 °C. After calibration, the viscosity of the sample was measured within the shear rate range of 0.1–500 s^−1^.

### 2.4. Stability of PSMs

#### 2.4.1. Appearance of PSMs

In order to investigate the storage stability of pumpkin seed milk, 10 mL of each freshly prepared PSM was accurately transferred into the glass bottle with a lid and placed at room temperature for 30 days. Photographs were taken at 1, 3, 7, 14, 21, and 30 days to observe the phase separation of the different PSMs.

#### 2.4.2. Centrifugal Stability of PSMs

Samples with a certain mass were accurately weighed into the centrifuge tubes and centrifuged at 9030× *g* for 20 min at 4 °C with a CR30NX high-speed centrifuge (5840R, Koki Holdings Co. Ltd., Tokyo, Japan). After centrifugation, the supernatants were removed, the precipitation mass was weighed, and the centrifugation precipitation rate was calculated. The smaller the precipitation rate, the better the stability of the milk [28]. The centrifugation precipitation rate was calculated with the following formula:Centrifugation precipitation rate = (m_2_ − m)/(m_1_ − m) ×100%(1)
where m is the mass of the empty centrifuge tube, m_1_ is the total mass before centrifugation, and m_2_ is the total mass after centrifugation.

#### 2.4.3. Effect of Salt Concentration on Stability of PSMs

Freshly prepared pumpkin seed milks were diluted with the same volume of concentrated salt (NaCl) solution to form the final sample with the salt concentrations of 0.1, 0.2, 0.3, 0.4, and 0.5 mol/L [29]. After the samples were diluted 100 times, the particle sizes and ζ-potentials were measured by the multi-angle particle size and high-sensitivity ζ-potential analyzer (DLS, Zetasizer Nano, Malvern Instruments Ltd., Worcestershire, UK) to evaluate the stability. 

#### 2.4.4. Effect of Freeze–Thaw on Stability of PSMs

Here, 5 mL of each freshly prepared pumpkin seed milk sample was placed in the freezer (−20 °C) and frozen for 24 h. After 24 h, the samples were taken out and placed in a water bath (37 °C) until half the ice was melted, and then transferred to a refrigerator to allow complete thawing. This was regarded as one complete freeze–thaw cycle. Each sample was subjected to three freeze–thaw cycles. After that, samples were diluted 100 times, and the particle size and ζ-potential were measured.

#### 2.4.5. Effect of Heat Treatment on Stability of PSMs

Here, 5 mL of each freshly prepared pumpkin seed milk sample was put in a test tube and heated in a water bath at 30 °C, 60 °C, and 90 °C for 6 h, then cooled to room temperature and diluted 100 times. The particle size and ζ-potential were measured by a multi-angle particle size analyzer and a highly sensitive ζ-potential analyzer.

### 2.5. Statistical Analysis

The results are presented as mean ± standard deviation (SD) of three technical replicates. The significant differences between the means were evaluated by ANOVA using IBM SPSS Statistics 26.0 and values of *p* < 0.05 were considered statistically significant.

## 3. Results and Discussion

### 3.1. Characterizations of Pumpkin Seed Milk

#### 3.1.1. Confocal Laser Scanning Microscope (CLSM) Observation

Figure 1 shows the confocal laser scanning microscope images of pumpkin milk droplets made from pumpkin seeds roasted at different temperatures. The red image represents protein in PSMs upon being dyed with Nile red. As shown in Figure 1, the pretreatment temperature had a significant influence on the flocculation and coalescence of pumpkin seed milk. The fluorescent images of RAW and PSM120 were similar and both were aggregated. PSM200 had relatively scattered droplets. This may be due to an increase in soluble protein in milk as the pretreatment temperature increased. At higher protein concentrations, more emulsifiers can be used to cover the oil–water interface, resulting in smaller droplet sizes. Yun et al. [30] also found that the quantity and concentration of adsorbed proteins at the interface of milk treated with preheating were significantly increased, thus showing good anti-flocculation stability and oxidation stability. The results of CLSM showed that high-temperature roasting could make the pumpkin seed milk droplets disperse, so that the distribution of pumpkin seed milk was more uniform and the stability was improved.

#### 3.1.2. Scanning Electron Microscope (SEM)

Scanning electron microscope images of freeze-dried powders of RAW, PSM120, PSM160, and PSM200 are shown in Figure 2. The morphologies of freeze-dried PSMs were significantly different. The surface of freeze-dried PSM120 was smooth and dense, and mushroom-like. While PSM160 showed a regular spherical distribution, PSM200 was spongy and porous, and RAW had a denser structure. These results were in accordance with CLSM results. This may also be explained by the increase of soluble proteins that served as emulsifiers during the preparation of PSMs, which will lead to a smaller droplet size and increased electrostatic repulsion between the droplets [31]. Yan et al. [32] found that roasting changed the structure of cashew protein, increased the development degree of protein molecules, exposed hydrophobic groups, and increased hydrophobic interactions, forming a three-dimensional network structure, which will eventually affect its application as an emulsifier. The SEM results showed that baking made the structure of PSM loose and porous, the interaction force between protein particles and fat pellets increased, and the particle size of PSM decreased. 

#### 3.1.3. Particle Size Distribution

Figure 3 shows the particle size distribution of PSMs at 2 h and after being stored for 7 days. As shown in Figure 3, the particle size distribution curve of pumpkin seed milk prepared by different roasting temperatures was different. Compared with 2 h, the particle size of PSMs increased after 7 days of storage. The results showed that the particle size of protein molecules increased with the aggregation of protein particles after storage. Particle size distribution and average particle size were important indexes to evaluate milk stability [33]. With the increase in the pre-roasting temperature of the raw material, the particle size of PSMs decreased. Again, this may be caused by increased protein in PSM prepared with pumpkin seeds roasted at higher temperatures. Another possible reason is the inactivation of lipoxygenase. Studies showed that lipoxygenase can promote the formation of protein aggregates through disulfide and non-covalent bonds between protein molecules, thus increasing the average particle size of milk [34]. Increased pretreatment can cause the inactivation of lipoxygenase and prevent protein aggregation. This indicates that with the extension of storage time, the particle size increases due to aggregation among PSM molecules, and the stability of PSM deteriorates after a long storage time.

#### 3.1.4. Rheological Property

As shown in Figure 4, thermal pretreatment of raw material significantly affected the rheological behavior of pumpkin seed milk. At the low shearing rate, PSMs showed shear-thinning behavior, i.e., the apparent viscosity decreased as the shearing rate increased. For example, the apparent viscosity of PSM200 decreased from 8.90 to 1.25 mPa·s when the shearing rate increased from 0 to 50 s^−1^. This was due to the unstable flow of the liquid when measuring viscosity at lower shearing rates. Then, the viscosity increased slowly with the increase of the shearing rate, showing a shear-thickening behavior. For example, the apparent viscosity of PSM200 increased from 1.25 to 1.73 mPa·s when the shearing rate increased from 50 to 500 s^−1^. Overall, PSM120, PSM160, and PSM200 showed higher viscosity than raw PSM [35]. After heating, high-temperature denaturation of protein and clustering resulted in the increase of sample viscosity and the decrease of the gel effect. Increased viscosity impedes droplet movement due to gravity or Brownian motion and inhibits further fusion and coalescence between proteins [33,36]. This was consistent with the conclusion of Bernat et al. [33], who found that the viscosity of heat-treated almond products was higher than that of hazelnut milk. The results showed that the viscosity of PSM increased with the increasing roasting temperature, but there was no positive correlation between the viscosity and the baking temperature. The increased viscosity further promotes the physical stability of PSM.

### 3.2. Stability of Pumpkin Seed Milk

#### 3.2.1. Storage Stability

To study the stability of PSMs, pumpkin seed milks were stored at room temperature in the dark for 30 days, and photos were taken on days 1, 3, 7, 14, 21, and 30. As shown in Figure 5, after 7 days of storage, samples of RAW and PSM120 began to sediment. PSM160 began to sediment after 14 days, while PSM200 did not separate within 30 days. Compared with RAW and PSM120, the storage stability of PSM160 and PSM200 was greatly improved. Dai et al. [37] reported that the hydrophobic groups inside peanut proteins were exposed after heat treatment, and the surface electrostatic charge was increased. Hydrophobic bonds and disulfide bonds of protein particles formed a gel, and the water-holding capacity and oil-mixing capacity of the emulsion were enhanced. After the surface of the oil droplets was adsorbed by proteins, the upward movement speed became smaller, which slowed down the phase separation and improved the stability [31].

#### 3.2.2. Centrifugal Stability

In this study, the centrifugal precipitation rate was used to characterize the centrifugal stability of pumpkin seed milk. The physical instability of pumpkin seed milk was mainly manifested in fat floating and protein sinking. The stability of pumpkin seed milk was improved with the decrease in the centrifugal precipitation rate. As shown in Figure 6, the centrifugal precipitation rates of RAW and PSM200 were significantly different. The centrifugal precipitation rate of unroasted pumpkin seed milk was higher than that of roasted pumpkin seed milk. PSM200 had the lowest centrifugal precipitation rate, which showed the best stability. This indicated that the stability of pumpkin seed milk was improved by roasting pretreatment, and the improvement was more obvious with the increase in roasting temperature. The stability of pumpkin seeds was not only affected by particles, but also by the interaction of composition, viscosity, and microstructure [28]. Roasting improves the hydration capacity and surface hydrophobicity of proteins, increases the interaction between molecules, forms a network gel structure between proteins, and increases the viscous resistance of particle deposition [33]. Therefore, increasing the roasting temperature can improve the centrifugal stability of pumpkin seed milk.

#### 3.2.3. Effect of Salt Concentration on Average Particle Size and ζ-Potential of Roasted Pumpkin Seed Milk

##### Effect of Salt Concentration on Average Particle Size of PSMs

Figure 7a shows the average particle size of roasted pumpkin seed milk at different NaCl concentrations. As the NaCl concentration increased, the particle size first increased and then decreased, with a maximum at a NaCl concentration of 0.3 mol/L. On the other hand, as the roasting temperature increased, the particle size decreased, with the lowest particle size being observed for PSM200 (roasted at 200 °C). Compared with the similar experimental results of Minmin et al. [38], the influence of salt concentration on the average particle size of PSMs was not significant, which indicated that the pumpkin seed milk prepared by roasting pumpkin seeds had better ion stability. The addition of salt will remove the charge on the protein surface, leading to the reduction of electrostatic repulsion between droplets and the highly flocculated state of the emulsion, resulting in the increase of the particle size. However, with the further electrostatic shielding effect, the emulsion will turn from flocculation to condensation, so the particle size will increase first and then decrease [37].

##### Effect of Salt Concentration on ζ-Potential of PSMs

As can be seen from Figure 7b, the ζ-potential value of PSMs ranged from −5 mV to 55 mV under different salt concentrations. When the pH of pumpkin seed milk was higher than the isoelectric point, the protein was negatively charged [33]. With the increase of NaCl concentration, the ζ-potential of pumpkin seed milk decreased first and then increased. The ζ-potential of PSM120 and PSM200 decreased first, then increased, and then decreased again. There was no significant correlation between PSM160 and salt concentration. At the same NaCl concentration, the absolute values of RAW charge were all smaller than PSM120 and PSM160 and had no significant correlation with PSM200. The results indicated that the addition of Na^+^ can prevent the emulsification of pumpkin seed milk to a certain extent, so that pumpkin seed milk had a higher ionic stability [29].

#### 3.2.4. Effect of Freeze–Thaw Cycles on the Stability of PSMs

##### Effect of Freeze–Thaw Cycles on the Average Particle Size of PSMs

In the freeze–thaw process, the crystallization of oil and water would cause the separation of oil and water in milk, affecting its stability, as shown in Figure 8a. At the same temperature, the average particle size of PSMs increased with the increase of the freeze–thaw times, but there was no significant effect on the average particle size of PSM200 during the freeze–thaw process. With the same number of freeze–thaw cycles, the average particle size was decreased by increasing the roasting temperature. With the increase in freeze–thaw times, the particle size significantly changed due to the rearrangement of the protein particle network and the aggregation of fat spheres in the microstructure [39]. When the seeds were roasted at 200 °C, the protein denaturation was serious, the exposure of hydrophobic groups increased, and the protein particles were closely bound to the fat so that they maintained a good shape of milk particles in the freeze–thaw process and reduced the aggregation [40,41]. Therefore, the particle size of PSM200 did not change significantly after freeze–thaw cycles. The results showed that the pumpkin seed milk prepared by roasting at high temperatures could better maintain the droplet structure and had good freeze–thaw stability.

##### Effect of Freeze–Thaw Cycles on ζ-Potential of PSMs

In Figure 8b, the ζ-potential of pumpkin seed milk varied from −15 mV to 55 mV under different salt concentrations. The repulsive force between particles increased with the increase of the absolute value of ζ-potential, and thus the stability of the emulsion system was improved with the decrease in the possibility of particle polymerization [41]. It can be seen from the figure that the absolute value of ζ-potential of the 4 kinds of pumpkin seed milk was greater than 30 when there was no freeze–thaw cycle. With the increase of freeze–thaw times, the absolute value of ζ-potential of the four kinds of pumpkin seed milk decreased, indicating that the freeze–thaw cycle would destroy the stability of pumpkin seed milk.

#### 3.2.5. Effect of Heat Treatment on the Stability of PSMs

##### Effect of Heat Treatment on the Average Particle Size of PSMs

As can be seen from Figure 9a, heat treatment at a high temperature of 200 °C did not have a significant effect on the particle size of pumpkin seed milk (PSM200). With the increase of heat treatment temperature, the particle size of RAW and PSM120 obviously increased, indicating that the particles had gathered. The interaction between proteins can be attributed to thermal denaturation and aggregation enhancement. Under the hydrophobic interaction, the surface hydrophobicity was enhanced, and the aggregate size was larger. Similar results have been observed in the case of peanut milk, where roasting significantly increased the particle size and improved the resistance of peanut milk to sterilization heat treatment. This is believed to be due to the Maillard reaction and its effect on the conformation and properties of proteins [42,43].

##### Effect of Heat Treatment on ζ-Potential of PSMs

The effect of heat treatment on the ζ-potential of PSMs is shown in Figure 9b. As the heat treatment temperature was increased, the particle size of PSMs was increased, and the absolute value of their ζ-potential was decreased. This indicated that the electrostatic repulsion was weakened by the reduction of the molecular surface charge, resulting in protein aggregation and poor stability [44]. At 90 °C, the large negative ζ-potential of PSMs was less than 30. The influence of heat treatment on absolute ζ-potential was intensified with the increase in the pretreatment temperature. Zhang et al. [45] believed that heating changed the interface properties, exposed the hydrophobic groups of proteins, increased the electrostatic attraction, or formed disulfide bonds, and thus increased the electrostatic attraction between droplets, leading to the aggregation of milk. 

## 4. Conclusions

In this study, we investigated the effects of roasting on the physical and chemical properties of pumpkin seed milk. The pumpkin seeds were roasted at temperatures of 120 °C, 160 °C, and 200 °C, and the resulting pumpkin seed milks were compared to the milk made from unroasted pumpkin seeds. Our results showed that higher roasting temperatures led to a decrease in particle size, and an increased stability of PSMs compared to the unroasted pumpkin seed milk. The effects of salt concentration, freezing and thawing cycles, and heat treatment on the stability of PSM200 were minimal but had a significant influence on RAW. In conclusion, our results suggested that thermal pretreatment of raw materials may enhance the physiochemical properties and stabilities of pumpkin seed milk. In future studies, we aim to determine the protein content, structure, properties, as well as lipoxygenase content in each PSM sample, to gain a deeper understanding of mechanisms underlying these phenomena. The result of this study may provide important implications for the development of pumpkin seed milk and other plant milk products.

## Figures and Tables

**Figure 1 foods-12-01056-f001:**
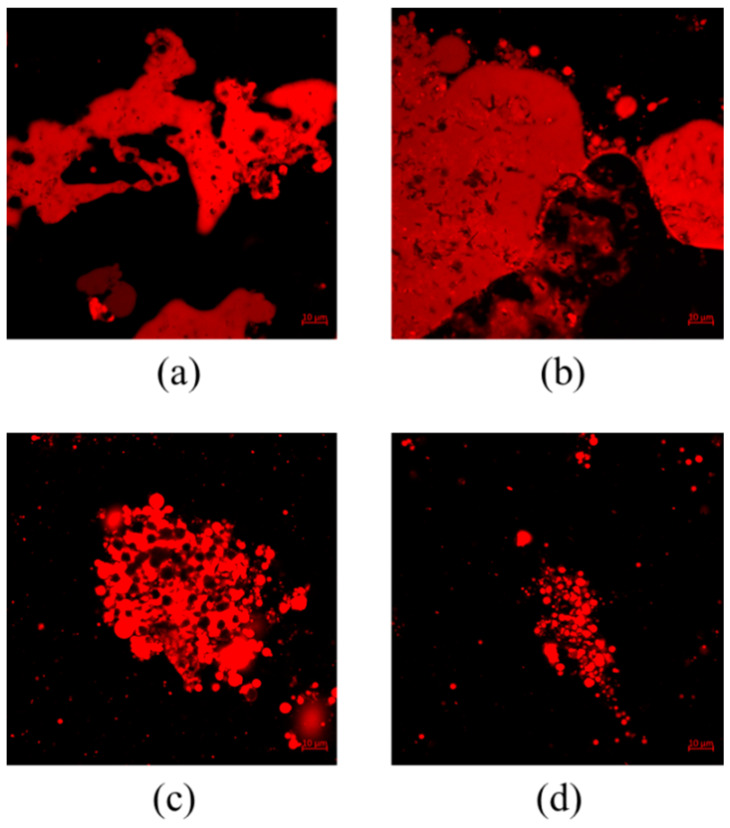
CLSM images of pumpkin seed milk droplets prepared with different pumpkin seeds: (**a**) raw pumpkin seeds, (**b**) pumpkin seeds roasted at 120 °C, (**c**) pumpkin seeds roasted at 160 °C, and (**d**) pumpkin seeds roasted at 200 °C.

**Figure 2 foods-12-01056-f002:**
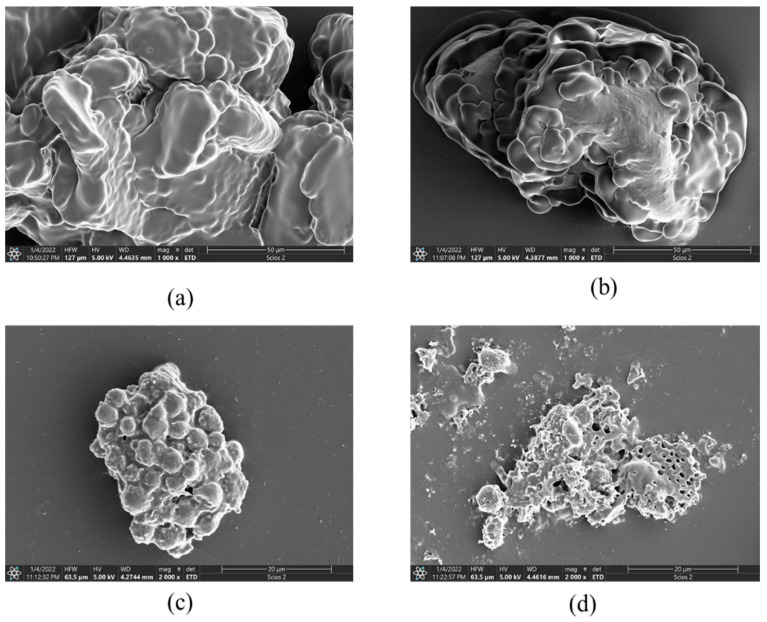
SEM images of freeze-dried pumpkin seed milk prepared with different pumpkin seeds. (**a**) RAW: raw pumpkin seeds, (**b**) PSM120: pumpkin seeds roasted at 120 °C, (**c**) PSM160: pumpkin seeds roasted at 160 °C, and (**d**) PSM200: pumpkin seeds roasted at 200 °C.

**Figure 3 foods-12-01056-f003:**
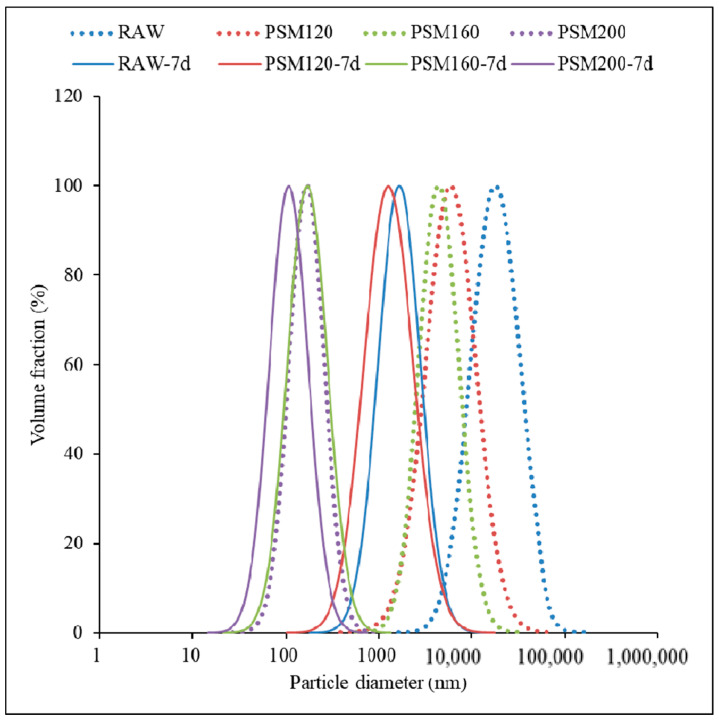
Particle size of pumpkin seed milk prepared with different pumpkin seeds. RAW: raw pumpkin seeds, PSM120: pumpkin seeds roasted at 120 °C, PSM160: pumpkin seeds roasted at 160 °C, and PSM200: pumpkin seeds roasted at 200 °C. Solid line: fresh pumpkin milk; dotted line: pumpkin milk stored at 20 °C for 7 days.

**Figure 4 foods-12-01056-f004:**
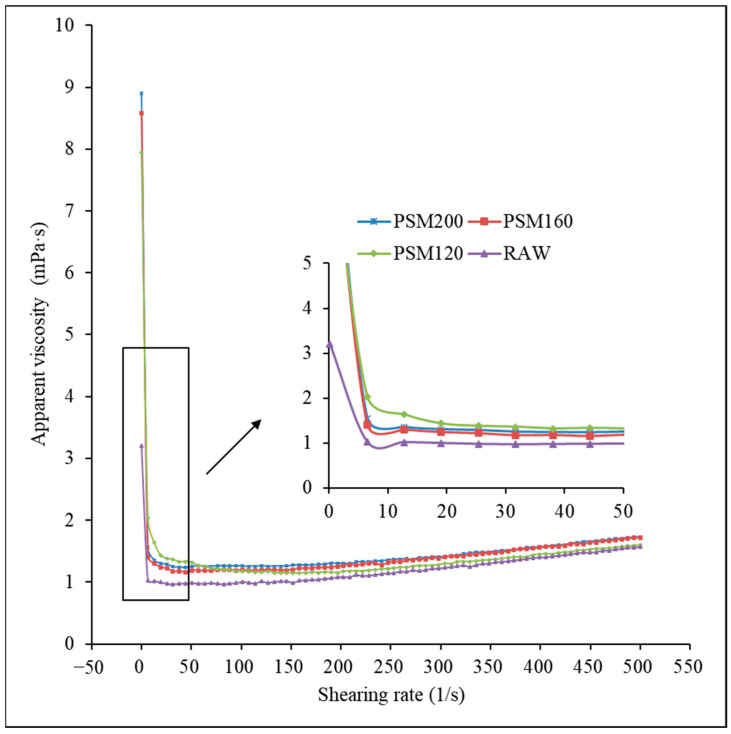
Rheology of pumpkin seed milk prepared with different pumpkin seeds. RAW: raw pumpkin seeds, PSM120: pumpkin seeds roasted at 120 °C, PSM160: pumpkin seeds roasted at 160 °C, and PSM200: pumpkin seeds roasted at 200 °C.

**Figure 5 foods-12-01056-f005:**
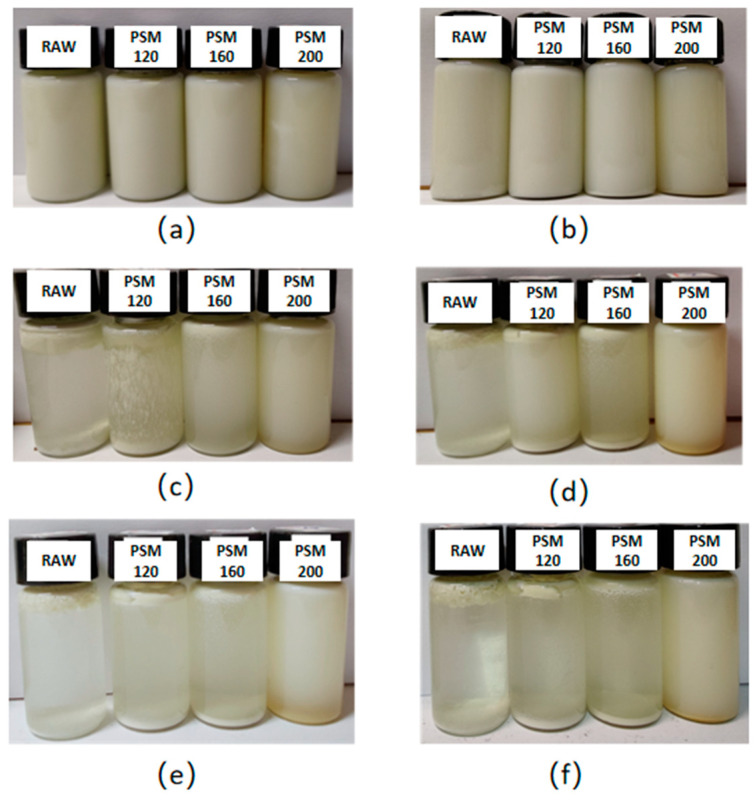
Appearance of PSM after storage for different times. (**a**–**f**) Storage for 1, 3, 7, 14, 21, and 30 days, respectively. RAW: raw pumpkin seeds, PSM120: pumpkin seeds roasted at 120 °C, PSM160: pumpkin seeds roasted at 160 °C, and PSM200: pumpkin seeds roasted at 200 °C.

**Figure 6 foods-12-01056-f006:**
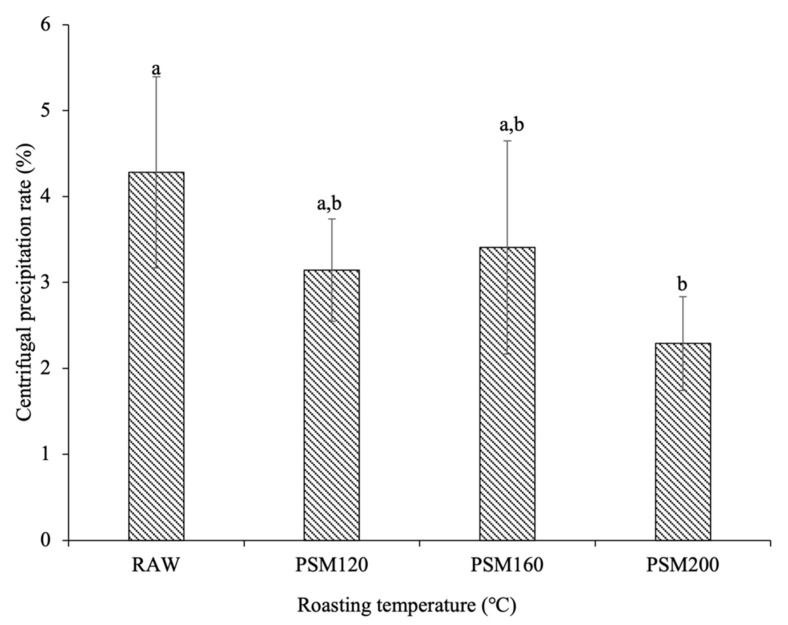
Centrifugal precipitation rate of pumpkin seed milk prepared with different pumpkin seeds. RAW: raw pumpkin seeds, PSM120: pumpkin seeds roasted at 120 °C, PSM160: pumpkin seeds roasted at 160 °C, and PSM200: pumpkin seeds roasted at 200 °C. Different lowercase letters represent significant differences (*p* < 0.05).

**Figure 7 foods-12-01056-f007:**
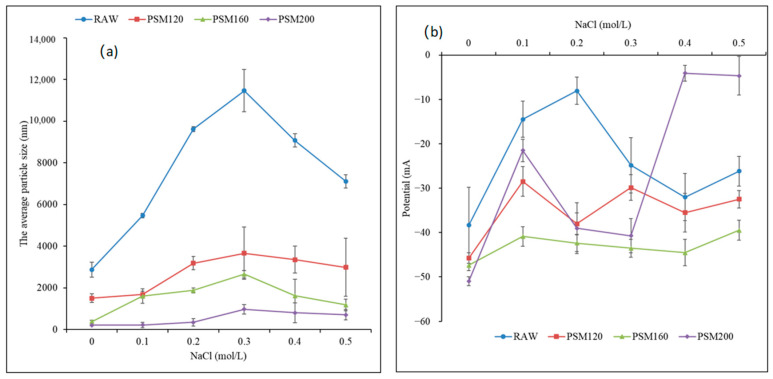
(**a**) Effect of salt concentration on average particle size of PSMs. (**b**) Effect of salt concentration on ζ-potential of PSMs. RAW: raw pumpkin seeds, PSM120: pumpkin seeds roasted at 120 °C, PSM160: pumpkin seeds roasted at 160 °C, and PSM200: pumpkin seeds roasted at 200 °C. Different lowercase letters represent the significant difference of the ζ-potential of different pumpkin seed milks at the same salt concentration, and different uppercase letters represent the significant difference of ζ-potential of the same milk at different salt concentrations.

**Figure 8 foods-12-01056-f008:**
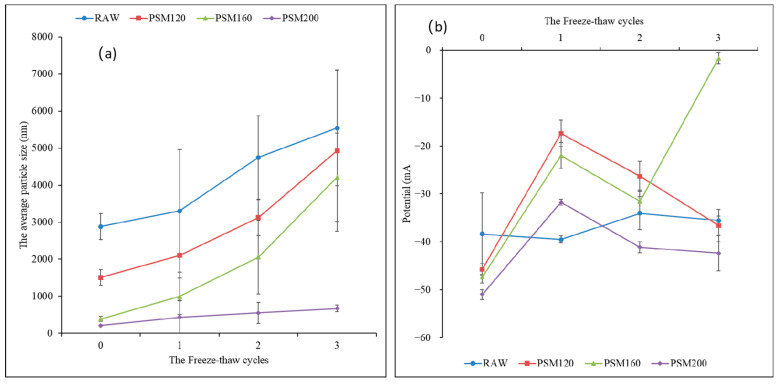
(**a**) Effect of freeze–thaw cycles on average particle size of PSMs. (**b**) Effect of freeze–thaw cycles on ζ-potential of PSMs. RAW: raw pumpkin seeds, PSM120: pumpkin seeds roasted at 120 °C, PSM160: pumpkin seeds roasted at 160 °C, and PSM200: pumpkin seeds roasted at 200 °C. Different lowercase letters represent the significant difference of ζ-potential of different pumpkin seed milks at the same salt concentration, and different uppercase letters represent the significant difference of ζ-potential of the same milk at different salt concentrations.

**Figure 9 foods-12-01056-f009:**
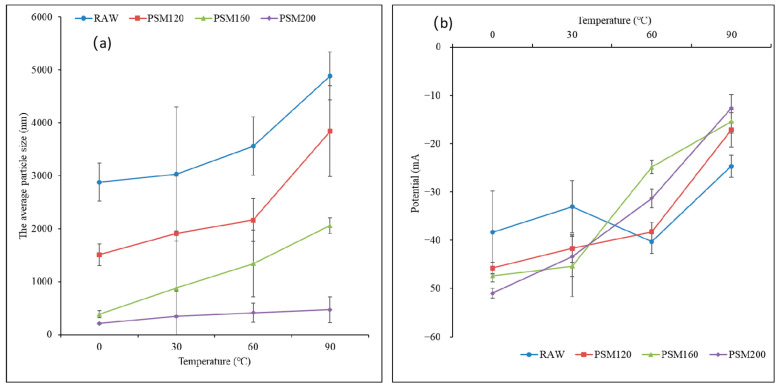
(**a**) Effect of heat treatment on average particle size of PSMs. (**b**) Effect of heat treatment on ζ-potential of PSMs. RAW: raw pumpkin seeds, PSM120: pumpkin seeds roasted at 120 °C, PSM160: pumpkin seeds roasted at 160 °C, and PSM200: pumpkin seeds roasted at 200 °C. Different lowercase letters represent the significant difference of ζ-potential of different pumpkin seed milks at the same salt concentration, and different uppercase letters represent the significant difference of ζ-potential of the same milk at different salt concentrations.

## Data Availability

The data used to support the findings of this study can be made available by the corresponding author upon request.

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
