# Peer review of "Effect of Thermal Pretreatment on the Physiochemical Properties and Stability of Pumpkin Seed Milk"

_foods, 2023, doi:10.3390/foods12051056_

Round 1
Reviewer 1 Report
Effect of Thermal Pretreatment on the Physiochemical Properties and Stability of Pumpkin Seed Milk
Overall write-up is good
In reference section, page numbers are missing in some cases
In problem statement, there must be emphasis on minimizing reduction of waste and preparation of cost effective plant based milk alternative to overcome the nutritional deficiencies of suffering population
Higher temperature no doubt improves stability of final product but it leads to loss of valuable nutrients, we can use the approach of UHT employing higher temperature but for short time like in seconds
For future recommendations, there should be focus on exploring its nutraceutical potential to commercialize as a plant based milk
Secondly its clinical trials should be conducted as you have mentioned that it has the ability to counteract hypertension, diabetes and anti-tumor and possess phenolic compounds
It should be added in the form of extract or powder form for nutraceutical purpose.
Other comments:
Line 30. Please add these latest citations for plant based:
Development and storage stability of chickpea, mung bean and peanut-based ready to use therapeutic food (RUTF) to tackle Protein-energy malnutrition.
Development of imitated meat product by utilizing pea and lentil protein isolates.
32. lease give space before the reference. This comment will be for whole manusctript.
44. citation is not right.
106 and 114. Why bold, no subheading.
Author Response
请参阅附件

Reviewer 2 Report
The MS requires extensive revision by professional help as it was difficult to read. The research design is too simple and the study did not include prior research on PSM. The references need some attention. I added many comments for the authors to consider.

Author Response
Please find the author's response in the attachment.

Reviewer 3 Report
Comments
The general question is whether the physico-chemical advantages gained by increasing the temperature of the heat treatment do not obscure the objective of obtaining a product with high nutritional value.
The choice of roasting temperatures for pumpkin seeds, i.e. 120, 160 and 200°C, and the duration of the process (i.e. 10 min), was dictated by what. Do the values of these parameters relate to similar studies conducted in this area. Such high temperatures and the heat treatment time used can lead to complete denaturation of the proteins and thus loss of their nutritional value, has this been tested.
Why was the suspension obtained percolated through gauze immediately after blending and not after homogenisation, which is the last process of preparing the milks for testing.
Line 44. Pan[7] – should be Pan et al. [7] also for Line 109. Zhong [22] – check throughout the work
Line 179. Yun et al. [27] ; Line 289 Minmin [34], in References is different
Line 394. Antunes….. No. 1 missing
Author Response
请参阅附件

Reviewer 4 Report
Dear Authors,
I revised the paper “Effect of Thermal Pretreatment on the Physiochemical Properties and Stability of Pumpkin Seed Milk” with interest.
The paper is well written and presented. The experimental design is appropriate. Methods are sound. Results are clearly presented and discussed. Up-to-date references were included.
My main concern about the study is the lack of data on nutritional properties of the experimental milks. The paper investigated the physio-chemical properties of Pumpkin Seed Milks and the stability thereof. It was found that the higher is the roasting temperature of pumpkin seeds and the better are the physio-chemical properties of experimental milks. What about the nutritional properties? Please, mind that in the Introduction, you listed some advantages of plant-based milks over traditional animal milks. Hence, in my opinion, additional studies are needed to investigate the effect on heat treatments on the nutritional properties of the experimental milks. Moreover, in line 66-76, the nutritional properties of pumpkin seeds are listed, and the reader might expect the investigation of nutritional characteristic of pumpkin seed milks, as well.
If the nutritional properties are not investigated, why did you choose pumpkin seeds to formulate milk-like beverages?
Few minor comments:
Line 41: The sentence is not clear. Please, specify the two categories: i) raw material processing conditions; ii) ??
Line 49: please, add references.
Author Response
请参阅附件

Reviewer 5 Report
The manuscript shows interesting and practically valuable results related to roasting effects for producing pumpkin seed milk. Generally, the study was properly design and the manuscript was well prepared.
However, some revisions are suggested for improving the quality of the manuscript as shown below:
− Abstract: The sentences “The seeds were …to obtain pumpkin” should be shortened and combined in 1 sentence.
− Abstract, Line 15-19: the description of methods is too detailed. In the abstract, only the important methods should be concisely mentioned.
− Abstract: the summary of the main findings is too general; some important data should be provided to support the statements of findings.
− Line 107: “suing” should be changed to “using”
− Sections 1.1.1, 3.1.2, 3.1.3 and 3.1.4: A sentence showing the meaning of the finding (how the observed results may affect the properties of seed milk samples) should be made at the end of each section.
− Figure 7b, 8b and 9b should be changed to the line charts for the better visualization of the results
− Line 172: “Fig.1 was the confocal laser scanning microscope image…” should be changed to “Fig.1 shows the confocal laser scanning microscope images…”
− Line 219-222 and 236-238: the uses of (a) (b) (c) (d) should be removed as those are not related to the corresponding figures
− Line 254-255: “Storage for 1 d, 3 d, 7 d, 14 d, 21 d, 30 d.…” should be changed to “Storage for 1 day, 3 days, 7 days, 14 days, 21 days and 30 days, respectively.…”
− Line 244-245: The discussion “possibly due to the increased viscosity of the roasted pumpkin seed milk” is not reasonable as this is conflict with the result in section 3.1.4 which shows the viscosity of the PSM120 was the highest but not PSM200.
− Line 261-263: the description is incorrect as only the precipitation rates of RAW and PSM200 were significant different while those among 3 roasted samples or among RAW and PSM120 and PSM 160 were not significant.
Author Response
请参阅附件

Round 2
Reviewer 2 Report
The MS has improved substantially. There are a few language issues, but not major and probably can be picked up during the proofs process